# GAM-RAG: Gain-Adaptive Memory for Evolving Retrieval in Retrieval-Augmented Generation

Yifan Wang [* 1]  Mingxuan Jiang [* 1]  Zhihao Sun [1]  Yixin Cao [1]  Yicun Liu [1]  Keyang Chen [1]  Guangnan Ye [1]
Hongfeng Chai [1]

## Abstract

Retrieval-Augmented Generation (RAG) grounds large language models with external evidence, but many implementations rely on pre-built indices that remain static after construction. Related queries therefore repeat similar multi-hop traversal, increasing latency and compute. Motivated by *schema*-based learning in cognitive neuroscience, we propose GAM-RAG, a training-free framework that accumulates retrieval experience from recurring or related queries and updates retrieval memory over time. GAM-RAG builds a lightweight, relation-free hierarchical index whose links capture potential co-occurrence rather than fixed semantic relations. During inference, successful retrieval episodes provide sentence-level feedback, updating sentence memories so evidence useful for similar reasoning types becomes easier to activate later. To balance stability and adaptability under noisy feedback, we introduce an uncertainty-aware, *Kalman*-inspired gain rule that jointly updates memory states and uncertainty estimates. It applies fast updates for reliable novel signals and conservative refinement for stable or noisy memories. We provide a theoretical analysis of the update dynamics, and empirically show that GAM-RAG improves average performance by 3.95% over the strongest baseline and by 8.19% with 5-turn memory, while reducing inference cost by 61%.

## 1. Introduction

Retrieval-Augmented Generation (RAG) has established itself as the cornerstone for grounding Large Language Models (LLMs) in external, verifiable knowledge (Zhang et al.,

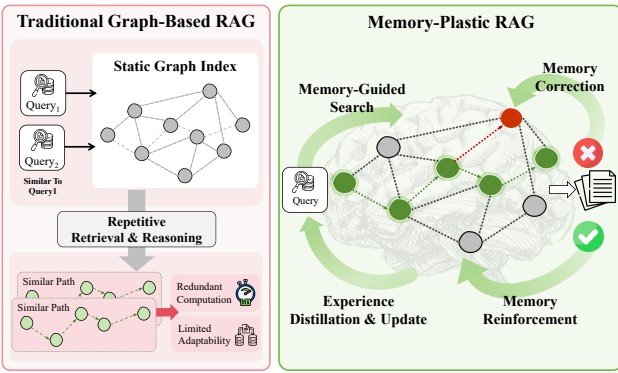

*Figure 1. Left*: Traditional graph-based RAG relies on a static, stateless graph index; related queries repeatedly traverse similar paths, resulting in redundant reasoning and limited adaptability. *Right*: Memory-plastic RAG distills retrieval feedback and performs memory updates, so future retrieval becomes more efficient and better aligned with evolving query needs.

2025; Sharma, 2025). Beyond simple semantic matching, recent graph-based RAG methods build structured indices that capture document connectivity and support multi-hop reasoning (Zhu et al., 2025b; Procko & Ochoa, 2024; Han et al., 2024; Luo et al., 2025). However, despite their structural differences, these existing paradigms share a fundamental limitation: they operate in a static and stateless manner (Peng et al., 2024b). Once the index is constructed, its topology remains fixed and does not incorporate feedback from historical retrievals (Fan et al., 2024). Consequently, as shown in Figure 1, each query is processed independently, and similar or recurring queries often trigger the same traversal and reasoning steps (Kashmira et al., 2025). This design prevents continual improvement from inference-time feedback, causing redundant reasoning and limiting adaptation to evolving query demands (Su et al., 2025).

We aim to introduce dynamically evolving memory into RAG to refine retrieval representations over time, without relying on expensive hand-crafted schemas. Inspired by Hebbian learning in neuroscience (Hebb, 2005), where synaptic connections strengthen through repeated co-activation ("cells that fire together, wire together"), we posit that a RAG system should exhibit structural plasticity, allowing

*Equal contribution  [1]Fudan University. Correspondence to: Guangnan Ye <yegn@fudan.edu.cn>.

*Proceedings of the 43rd International Conference on Machine Learning*, Seoul, South Korea. PMLR 306, 2026. Copyright 2026 by the author(s).

historical memory to guide retrieval for new events. Under this view, retrieval mirrors human cognitive evolution (Sekeres et al., 2024): as illustrated in Figure 1, it first establishes loose associations between entities and texts, and then reinforces reasoning paths that have proven effective in past inference, refining the entity-level memory representations. By continually accumulating and updating experience from recurring or related queries, a memory-guided RAG system can uncover implicit, task-specific logical shortcuts, improving both retrieval efficiency and accuracy and transforming static search into an adaptive process of cognitive updating.

To materialize this insight, we propose **GAM-RAG**, a training-free framework that optimizes retrieval through a gain-adaptive memory updating mechanism.

It starts from a lightweight structured index whose links capture potential co-occurrence, rather than fixed semantic relations. In particular, during inference, each retrieval episode provides sentence-level feedback. The system updates the corresponding node memories with query information so that evidence supporting similar reasoning patterns becomes easier to activate for future related queries. However, implementing such plasticity introduces a critical challenge: balancing stability and adaptability. Since retrieval feedback is noisy, indiscriminately updating memory with equal strength after each activation can accumulate spurious correlations. This can induce false associations and overfit to recent errors.

To address this issue, we introduce a *Kalman*-inspired Gain-Adaptive update mechanism. This method treats memory not merely as a storage of values, but as a dynamic state with an associated uncertainty. Our algorithm regulates memory updates by adjusting the learning gain according to memory reliability. It updates quickly for novel and high-confidence signals, while using smaller, damped updates for stable or noisy memories. This ensures that the system can rapidly absorb new knowledge without being dominated by recent noisy episodes or eroding reliable long-term structure. We further leverage the updated memories to actively guide subsequent retrieval. The refined memory states serve as query-aligned priors that modulate propagation and ranking weights, pulling supportive evidence closer in representation space. Theoretical analysis and extensive experiments across diverse settings demonstrate that GAM-RAG achieves consistent performance gains and reduced inference overhead as experience accumulates, validating the potential of training-free, evolutionary retrieval. Overall, our key contributions can be summarized as follows.

- We propose GAM-RAG, a gain-adaptive memory framework that accumulates retrieval experience from queries, allowing the retrieval organization to evolve over time for more effective evidence discovery.

- We develop an uncertainty-aware, *Kalman*-inspired update rule that jointly maintains memory states and their uncertainty estimates, enabling robust continual refinement under noisy retrieval feedback.

- We evaluate GAM-RAG theoretically and empirically across diverse settings. Compared with the strongest baseline, it improves average performance by 3.95%, and by 8.19% with 5-turn memorization, while reducing inference cost by 61%.

## 2. Related Work

Recent RAG systems increasingly encode semantic relations among textual entities using graph structures, enabling more effective multi-hop retrieval. Such graph structures provide an explicit substrate for chaining evidence across documents and hops, which can make retrieval more structured in multi-step settings. *KG-based RAG* explicitly represents these relations via knowledge graphs (KGs), over which multi-hop paths are composed to support reasoning (Peng et al., 2024a; Zhu et al., 2025a). In contrast, Hybrid RAG uses KG-derived links primarily as a cross-document index to steer passage retrieval, while keeping the evidence in raw text to preserve local context (e.g., ToG) (Cheng et al., 2025; Gutiérrez et al., 2025). Graph-based RAG structures multi-hop retrieval with an explicit graph, but this structure comes with costs. In practice, constructing and maintaining such graphs often depends on upstream extraction, linking, or schema choices, which affects coverage and upkeep. Graphs can be incomplete beyond the chosen ontology and expensive to build or maintain, which limits coverage and timely updates (Xu et al., 2024; Pan et al., 2024). Moreover, collapsing text into nodes and edges can obscure sentence-level provenance, and multi-hop traversal can quickly branch with unstable step-wise feedback (Xiang et al., 2025). As a result, fine-grained evidence attribution may be weakened, and errors can propagate across hops when branching and feedback are noisy.

While recent *Memory-augmented RAG* work introduces persistent, updateable state to improve retrieval efficiency. In this line, HippoRAG and HippoRAG 2 treat an external knowledge graph as non-parametric long-term memory: new observations can be written into the graph (often via LLM-based extraction), and queries are answered by projecting to the graph and reading out evidence through PageRank-style propagation (Jimenez Gutierrez et al., 2024; Gutiérrez et al., 2025). ReMindRAG moves further toward experience reuse by caching traversal traces as replayable path memory, so similar queries can reuse previously effective routes instead of re-exploring from scratch (Hu et al., 2025). Existing attempts to store experience as textual traces further suffer from growing context overhead and the absence of mechanisms to assess or adapt the reliability of

stored memories, motivating memory designs with explicit update control and uncertainty-aware reuse of evidence.

# 3. Methodology

In this section, we introduce GAM-RAG, a Gain-Adaptive Memory framework that continuously updates experiential knowledge without additional training, aiming to improve retrieval accuracy while reducing inference overhead. As illustrated in Figure 2, the GAM-RAG workflow consists of three stages: (a) graph construction and memory initialization. (b) memory-guided iterative retrieval. (c) uncertainty-aware dynamic memory updating.

## 3.1. Graph Construction and Memory Initialization

### 3.1.1. GRAPH CONSTRUCTION

Previous work has shown that constructing knowledge graphs (KGs) via explicit relation extraction is costly and the quality is often unreliable, which becomes even more severe for smaller models and resource-constrained deployments (Wu et al., 2025; Zhuang et al., 2025a). To avoid semantic loss introduced by relation-extraction errors while reducing indexing cost, we build semantic associations across texts via entity co-occurrence. Concretely, we apply a lightweight model spaCy (Ho et al., 2020) to perform sentence segmentation and named entity recognition (NER). Given the passage collection, we construct a hierarchical graph $\mathcal{G} = (\mathcal{V}, \mathcal{M})$ ordered by information granularity, where $\mathcal{V} = \mathcal{V}_\mathcal{E} \cup \mathcal{V}_\mathcal{S} \cup \mathcal{V}_\mathcal{P}$ consists of entity, sentence and passage nodes. In particular, the sentence–entity links can be represented by an incidence matrix:

$$\begin{aligned} \mathcal{M}_{ES} &= \left[M_{ij}\right]_{|\mathcal{V}_\mathcal{E}| \times |\mathcal{V}_\mathcal{S}|}, \quad M_{ij} = \mathbb{I}(e_i \in s_j), \\ \mathcal{M}_{SP} &= \left[M_{ij}\right]_{|\mathcal{V}_\mathcal{S}| \times |\mathcal{V}_\mathcal{P}|}, \quad M_{ij} = \mathbb{I}(s_i \in p_j), \end{aligned} \quad (1)$$

where $\mathcal{M}_{ES}$ encodes entity occurrence in sentences and $\mathcal{M}_{SP}$ encodes sentence containment in passages.

### 3.1.2. MEMORY INITIALIZATION

Supporting evidence for a query often appears in one or a few key sentences within long contexts. We choose sentences as the minimal memory unit, and assign the task-oriented memory vector $\mathbf{m}_i^{\text{task}} \in \mathbb{R}^d$ that summarizes its contribution to solving past queries. Notably, in time-sensitive tasks, purely semantic retrieval may fail to distinguish temporal differences, causing conflicts between evidence that is semantically similar but with different time constraints. To mitigate this, we further introduce a time-oriented memory vector $\mathbf{m}_i^{\text{time}} \in \mathbb{R}^d$ to represent the temporal coverage of a sentence. Correspondingly, we maintain two adaptive terms $\pi_i^{\text{task}}, \pi_i^{\text{time}} \in [0, 1]$ to modulate the reliability of these memories. For each sentence $s_i$, we initialize its memory state

as follows:

$$\text{Mem}(s_i) = \left(\mathbf{m}_i^{\text{task}}, \mathbf{m}_i^{\text{time}}, \pi_i^{\text{task}}, \pi_i^{\text{time}}\right), \quad (2)$$

where $\mathbf{m}_i^{\text{task}}$ is initialized as the sentence embedding, and $\mathbf{m}_i^{\text{time}} \in \mathbb{R}^d$ encodes the extracted time expression $\tau_i$ of $s_i$ using an LLM, if no temporal constraint is present, we set $\tau_i$ as NO_TIME. The uncertainty terms are initialized to 1.

## 3.2. Memory-Guided Iterative Retrieval

After graph construction and memory initialization, we design a memory-guided iterative retrieval procedure, enabling the system to leverage accumulated experience to localize evidence more quickly across iterations, improving both accuracy and recall. We decompose the retrieval process into: (1) initial entity activation, and (2) memory-guided iterative semantic propagation.

### 3.2.1. INITIAL ENTITY ACTIVATION

Given a query $Q$, we use spaCy to extract the query entity set $E_Q$. Then we prompt an LLM to extract the query's temporal constraint and encode it into $\mathbf{q}^{\text{time}} = \text{Enc}(\tau_Q)$ using a unified embedding model (all-mpnet-base-v2); if $Q$ contains no temporal description, we set it to NO_TIME and later disable the time memory in semantic propagation. For each extracted query entity, we align it to the most similar entity node in the graph as the first-round activation, using embedding similarity as the entity score:

$$\mathcal{A}_0 = \{(\tilde{e}_i, a_i)\}_{i=1}^{|E_Q|}, \quad \tilde{e}_i = \arg\max_{e \in \mathcal{V}_\mathcal{E}} \text{sim}\left(e_i^Q, e\right), \quad (3)$$

here $e_i^Q \in E_Q$, and $a_i$ denotes the cosine similarity score.

### 3.2.2. MEMORY-GUIDED SEMANTIC PROPAGATION

We perform multi-hop retrieval by iterating three steps: (i) entity-sentence score propagation, (ii) sentence-passage evidence aggregation, and (iii) sentence-entity re-activation. Throughout this process, sentence-level memory provides an experience-based prior: it re-weights entity-sentence propagation to favor previously validated and query-consistent evidence, and it further stabilizes exploration by down-weighting unreliable memories via gating.

**Entity-Sentence propagation.** Given the activated entities $\mathcal{A}_t$ at iteration $t$, we propagate their scores to connected sentences, with memory uncertainties controlling the gating sensitivity. For each sentence $s_i$, we compute the weight as:

$$\begin{aligned} g_i^{\text{task}} &= 1 + (1 - \pi_i^{\text{task}}) \cdot \cos\left(\mathbf{m}_i^{\text{task}}, \mathbf{q}\right), \\ g_i^{\text{time}} &= 1 + (1 - \pi_i^{\text{time}}) \cdot \cos\left(\mathbf{m}_i^{\text{time}}, \mathbf{q}^{\text{time}}\right), \quad (4) \\ w_i^{\text{sent}} &= s_i^{\text{sem}} \cdot g_i^{\text{task}} \cdot g_i^{\text{time}}, \end{aligned}$$

here $\mathbf{q}$ is the query embedding, $s_i^{\text{sem}}$ is the sentence–query cosine similarity. Let $\mathbf{a}_t \in \mathbb{R}^{|\mathcal{V}_\mathcal{E}|}$ denote the entity activa-

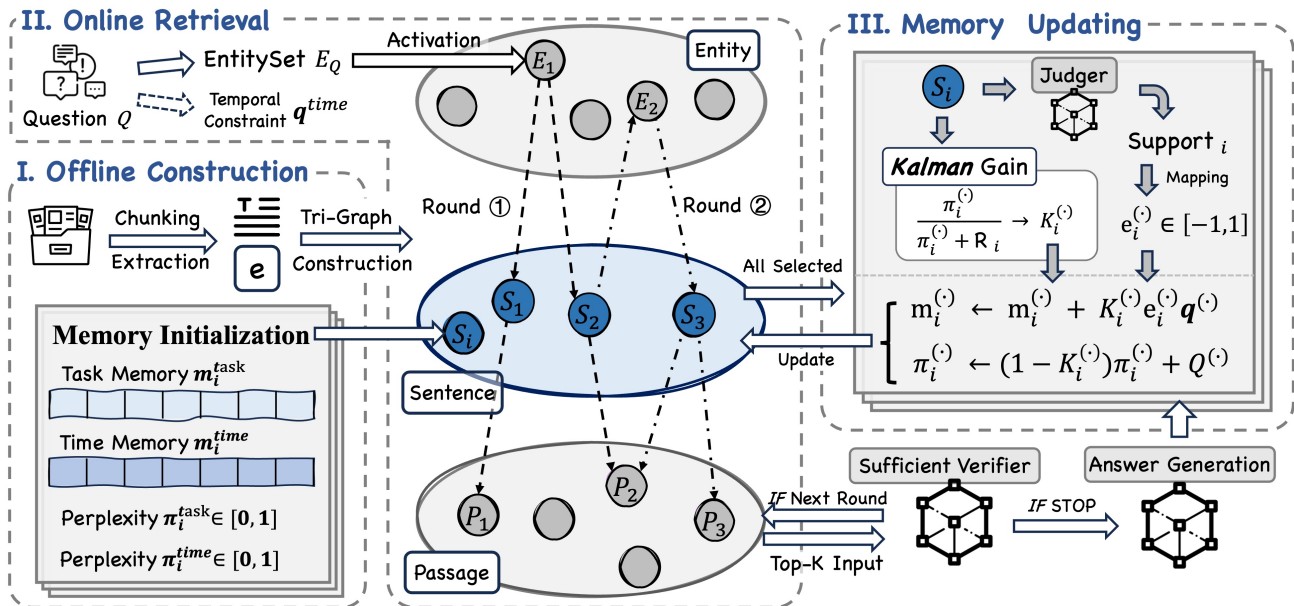

*Figure 2.* **Overview of our method.** GAM-RAG introduce a training-free, gain-adaptive sentence memory to retrieval. A query activates entity nodes and runs iterative graph propagation to discover multi-hop evidence. After each episode, a Kalman-inspired, uncertainty-aware gain update adjusts both memory states and their uncertainties to keep updates stable under noisy signals.

tion vector at iteration $t$ (i.e., $\mathbf{a}_t[j] = a_t(e_j)$ for activated entities). The entity-to-sentence propagated score is:

$$\mathbf{u}_t^{(S)} = \mathcal{M}_{ES}^\top \mathbf{a}_t, \qquad u_t(s_i) = \mathbf{u}_t^{(S)}[i]. \qquad (5)$$

Finally, we compute the normalized score by combining propagated entity scores with the sentence weight:

$$\text{score}_t(s_i) = \text{Norm}_1\Big(w_i^{\text{sent}} \cdot u_t(s_i)\Big), \qquad (6)$$

where $\text{Norm}_1(\cdot)$ performs $\ell_1$ normalization over all candidate sentences at iteration $t$, making the ranking invariant to global multiplicative scaling. This weighting explicitly accounts for multi-aspect agreement (semantics, experience, and temporal constraint). For entity-free queries, we directly propagate with the learned weights; similarly, potential entity ambiguity can be mitigated by the memory-alignment signals (e.g., $\cos(\mathbf{m}_i^{\text{task}}, \mathbf{q})$).

**Sentence-Passage evidence retrieval.** We select the top-$K_S$ sentences as the activated sentences at iteration $t$, and aggregate their contributions to passage nodes. Let $\mathbf{w}_t^{(S)} \in \mathbb{R}^{|\mathcal{V}_S|}$ be the sentence-weight vector where $\mathbf{w}_t^{(S)}[i] = \text{score}_t(s_i)$. We maintain a tiered accumulation of sentence evidence for each passage $p$:

$$\text{bonus}_t(p) = \big(\mathcal{M}_{SP}^\top \mathbf{w}_t^{(S)}\big)[p], \qquad (7)$$

which sums the t-tier contributions of activated sentences contained in $p$. To obtain a stable passage ranking across iterations, we combine a passage-level prior (passage–query

embedding similarity) with a log-accumulated bonus:

$$\text{score}_t(p) = \alpha \cdot \text{sim}(p, Q) + \sum_{t=1}^{T} \frac{\log\big(1 + \text{bonus}_t(p)\big)}{t}. \qquad (8)$$

Here $\alpha$ is a trade-off coefficient, the $\log(\cdot)$ with $1/t$ discount prevents a single tier from dominating.

**Sentence-Entity activation.** In multi-hop queries, a single propagation round may miss intermediate bridge entities. Thus, after ranking passages with Eq. 8, we provide the top-$K_S$ sentences to the LLM to verify whether the current evidence is sufficient to answer $Q$. If not, we trigger the next round by propagating scores back from activated sentences to entities. The activate entity score for the next iteration:

$$\mathbf{a}_{t+1} = \big(\mathcal{M}_{ES} \mathbf{w}_t^{(S)}\big) \oslash \big(\mathcal{M}_{ES} \mathbf{1}\big), \qquad (9)$$

where $\mathbf{a}_{t+1} \in \mathbb{R}^{|\mathcal{V}_\mathcal{E}|}$ is the sentence-to-entity propagated score, $\oslash$ denotes element-wise division, and $\mathbf{1}$ is an all-one vector, $(\mathcal{M}_{ES}\mathbf{1})[j] = |\mathcal{N}_S(e_j)|$ is the total number of sentence neighbors of entity $e_j$. This average aggregation ensures that an entity appearing in many generic sentences does not dominate, while an entity concentrated in a few high-scoring sentences can be strongly activated, leading to more reliable multi-hop exploration.

### 3.3. Uncertainty-aware Dynamic Memory Updating

Unlike prior approaches that rely on explicitly stored training samples or offline finetuning to inject experience (Li et al., 2024b; Zhou et al., 2025), GAM-RAG updates memory online in a training-free manner. Motivated by the

Kalman-filter view of *gain as a dynamic weight* (Welch & Bishop, 1995), we treat each sentence memory as a vector state and adjust it with an adaptive gain that depends on (i) the current uncertainty of the memory and (ii) the confidence of the LLM-as-judge signal. This yields two practical benefits: (1) *fast adaptation* when memory is still uncertain, and (2) *damped updates* once memory becomes stable, reducing overfitting to noisy retrieval episodes.

### 3.3.1. KALMAN-INSPIRED MEMORY UPDATE

After the retrieval iteration, we obtain a feedback signal $y_i \in \{-1, 1\}$ for selected sentences via an LLM judge, indicating whether sentence $s_i$ provides effective evidence for answering the query. For each sentence $s_i$, we maintain two memory states $\mathbf{m}_i^{\text{task}}, \mathbf{m}_i^{\text{time}} \in \mathbb{R}^d$ and their uncertainties $\pi_i^{\text{task}}, \pi_i^{\text{time}} \in [0, 1]$. Given the query embeddings $\mathbf{q}$ and $\mathbf{q}^{\text{time}}$, we update memory along the corresponding query direction (i.e., a 1D filter in the projected space).

**Residual.** We first compute the residual between the observation and the projected prediction. Intuitively, it quantifies the mismatch between the LLM-judged support and what the current memory predicts along the query direction:

$$e_i^{(\cdot)} = y_i - \hat{y}_i^{(\cdot)} = y_i - \cos\big(\mathbf{q}^{(\cdot)}, \mathbf{m}_i^{(\cdot)}\big), \qquad (10)$$

$e_i^{(\cdot)}$ indicates whether the memory should be pulled toward $(e_i^{(\cdot)} > 0)$ or pushed away $(e_i^{(\cdot)} < 0)$ from query $\mathbf{q}^{(\cdot)}$.

**Kalman gain with asymmetric observation noise.** We then compute a scalar Kalman gain, which acts as an adaptive learning rate of the memory:

$$K_i^{(\cdot)} = \frac{\pi_i^{(\cdot)}}{\pi_i^{(\cdot)} + R_i}, \qquad (11)$$

where $R_i$ is the observation-noise term that quantifies the uncertainty of the LLM judge. We use asymmetric noise levels $R_{\text{pos}}$ and $R_{\text{neg}}$. Here, positive support ($y_i = +1$) is triggered by the judge deeming the sentence supportive, and negative support ($y_i = -1$) is defined as "not marked as positive" and may still include hidden bridge evidence; to avoid missing such intermediate clues, we assign larger noise to negatives, making their updates more conservative.

**State update.** With the gain and residual, we update the memory state by a correction along the query direction:

$$\mathbf{m}_i^{(\cdot)} \leftarrow \mathbf{m}_i^{(\cdot)} + K_i^{(\cdot)} \cdot e_i^{(\cdot)} \cdot \mathbf{q}^{(\cdot)}. \qquad (12)$$

This update preserves the memory components orthogonal to $\mathbf{q}^{(\cdot)}$ and only adjusts the projected support degree, making it robust to unrelated semantic drift.

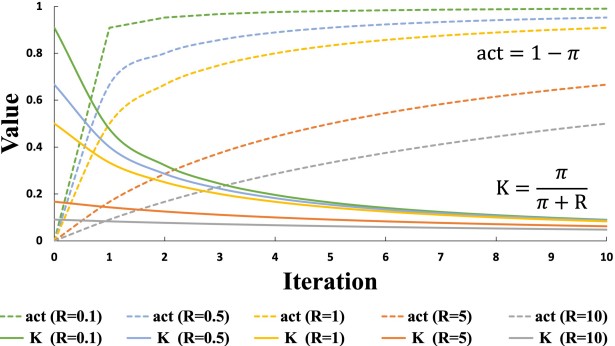

*Figure 3.* Kalman gain dynamics: under different observation-noise levels $R_i$, illustrating fast warm-up and damped refinement.

**Uncertainty update with process noise.** As the memory state becomes more reliable after the update, we accordingly reduce its uncertainty as:

$$\pi_i^{(\cdot)} \leftarrow \text{clip}_{[0,1]}\Big((1 - K_i^{(\cdot)})\pi_i^{(\cdot)} + Q^{(\cdot)}\Big), \qquad (13)$$

where $Q^{(\cdot)} > 0$ is a process-noise offset that prevents $\pi_i^{(\cdot)}$ from collapsing to 0 after many updates. For non-time-sensitive queries (i.e., $\mathbf{q}^{\text{time}} = \text{NO\_TIME}$), we disable time updates by setting $K_i^{\text{time}} = 0$.

### 3.3.2. DYNAMICS OF VARIABLES

Below we summarize how key variables evolve and the resulting behaviors. A detailed theoretical analysis of the update effectiveness is provided in Appendix A.

**Fast warm-up vs. damped refinement.** The Kalman gain $K_i^{(\cdot)}$ acts as an effective learning rate for the state update in Eq. 12. As illustrated in Fig. 3, when a sentence memory is still uncertain (large $\pi_i^{(\cdot)}$), we have $K_i^{(\cdot)} \approx 1$, leading to large, fast updates (*warm-up*). As updates accumulate, Eq. 13 reduces $\pi_i^{(\cdot)}$, which decreases $K_i^{(\cdot)}$ and naturally damps the update magnitude, making memory increasingly stable and less sensitive to any single retrieval episode. Meanwhile, the judge-noise term $R_i$ controls how conservative updates are: smaller $R_i$ yields a larger gain and faster adaptation, while larger $R_i$ shrinks the gain and slows updates, improving robustness to noisy feedback.

## 4. Experiment

### 4.1. Experimental Settings

**Datasets.** We evaluate GAM-RAG under diverse task settings to assess both effectiveness and efficiency, covering multi-hop QA, time-sensitive QA, and domain-specific QA. (1) **Multi-hop QA.** We use three widely-adopted multi-hop benchmarks: 2WikiMultiHopQA (2Wiki) (Ho et al., 2020), HotpotQA (Yang et al., 2018), and MuSiQue (Trivedi et al.,

*Table 1.* GPT-Acc and Contain-Acc. on five QA benchmarks. The best result is in bold and the second-best is underlined. w/k-turn memorization reports multi-round evaluation on the same dataset; the superscript ↑ percentages denote improvements relative to 0-turn memorization.

| Methods | 2Wiki | | HotpotQA | | MuSiQue | | TimeQA | | Medical | |
|---|---|---|---|---|---|---|---|---|---|---|
| | GPT-Acc | Contain-Acc. | GPT-Acc | Contain-Acc. | GPT-Acc | Contain-Acc. | GPT-Acc | Contain-Acc. | GPT-Acc | F1. |
| **Standard LLM** | | | | | | | | | | |
| Direct Answering | 30.20 | 33.70 | 40.00 | 37.60 | 14.90 | 13.20 | 30.18 | 31.85 | 42.10 | 32.16 |
| Chain-of-Thought (CoT) | 31.40 | 34.50 | 39.80 | 38.20 | 15.80 | 13.60 | 30.62 | 33.67 | 42.52 | 34.50 |
| **Vanilla RAG** | | | | | | | | | | |
| GritLM / GritLM-7B | 42.10 | 44.80 | 55.20 | 51.30 | 28.50 | 25.80 | 41.81 | 42.68 | 58.83 | 54.21 |
| nvidia/NV-Embed-v2 | 44.60 | 47.30 | 58.90 | 56.80 | 28.90 | 26.10 | 43.89 | 44.35 | 61.44 | 56.27 |
| all-mpnet-base-v2 | 43.50 | 48.20 | 58.40 | 56.20 | 29.60 | 26.80 | 41.29 | 44.14 | 60.96 | 55.30 |
| **GraphRAG-style methods** | | | | | | | | | | |
| GFM-RAG | 59.80 | 65.50 | 65.60 | 61.80 | 34.20 | 28.70 | 42.83 | 44.79 | 56.07 | 50.24 |
| HippoRAG2 | 55.70 | 61.90 | 64.30 | 62.90 | 24.80 | 27.30 | 41.96 | 43.89 | 60.77 | 52.48 |
| LinearRAG | 61.70 | 69.20 | 66.10 | 63.30 | 37.20 | 32.50 | 44.62 | 47.82 | 63.72 | 54.92 |
| PoG | 54.10 | 60.70 | 58.10 | 60.90 | 31.60 | 29.20 | 41.72 | 43.06 | 61.40 | 53.76 |
| DyG-RAG | 40.70 | 41.80 | 51.20 | 48.30 | 20.40 | 17.80 | **51.26** | **52.38** | 51.28 | 44.92 |
| REMINDRAG | 61.40 | 68.20 | 74.20 | 71.30 | 35.30 | 30.80 | 43.89 | 45.72 | 64.25 | 56.22 |
| **GAM-RAG** | **64.20** | **71.80** | **74.90** | **72.50** | **43.30** | **38.00** | 49.45 | 51.35 | 65.47 | 58.98 |
| w/ 1-turn memorization | 65.30$^{\uparrow 1.7\%}$ | 72.80$^{\uparrow 1.4\%}$ | 74.90 | 71.90 | 44.50$^{\uparrow 2.8\%}$ | 39.20$^{\uparrow 3.2\%}$ | 49.12 | 51.12 | 66.72$^{\uparrow 1.9\%}$ | 61.92$^{\uparrow 4.7\%}$ |
| w/ 2-turn memorization | 67.80$^{\uparrow 5.6\%}$ | 74.60$^{\uparrow 3.9\%}$ | 76.20$^{\uparrow 1.9\%}$ | 74.30$^{\uparrow 2.5\%}$ | 45.80$^{\uparrow 5.8\%}$ | 41.30$^{\uparrow 8.7\%}$ | 50.32$^{\uparrow 1.8\%}$ | 52.36$^{\uparrow 2.0\%}$ | 68.81$^{\uparrow 5.1\%}$ | 63.23$^{\uparrow 7.2\%}$ |

2022), to evaluate complex multi-hop reasoning capability. (2) **Time-sensitive QA.** We adopt the hard split of TimeQA (Chen et al., 2021), which features implicit temporal expressions and cross-sentence reasoning, to evaluate robustness against temporal conflicts. (3) **Domain-specific QA.** We use the Medical dataset from GraphRAG-Bench (Xiao et al., 2025), containing substantial domain-specific knowledge, to evaluate generalization in specialized scenarios. Refer to Appendix B for details.

**Baselines.** We compare GAM-RAG against three groups of baselines. (1) **Standard LLM.** We evaluate direct answering and Chain-of-Thought (CoT) (Wei et al., 2022), which rely solely on the parametric knowledge of the underlying LLM. (2) **Vanilla RAG.** We implement a standard dense-retrieval RAG pipeline that retrieves by query-passage similarity. We use GritLM/GritLM-7B (Muennighoff et al., 2024), nvidia/NV-Embed-v2 (Lee et al., 2024), and all-mpnet-base-v2 (Song et al., 2020) as embedding models. (3) **GraphRAG-style methods.** We include representative state-of-the-art structured-retrieval baselines. Graph-search algorithm driven methods: GFM-RAG (Luo et al., 2025), HippoRAG2 (Gutiérrez et al., 2025), and LinearRAG (Zhuang et al., 2025b); LLM-guided traversal: PoG (Chen et al., 2024), which uses an LLM to plan exploration paths over the graph; Temporal-structure modeling for time-sensitive QA: DyG-RAG (Sun et al., 2025), which constructs an event graph to identify temporal anchors for retrieval. Experience-aware graph retrieval: REMINDRAG (Hu et al., 2025), which updates edge representations based on retrieval feedback.

**Metrics and implementations.** (1) Effectiveness: Following recent benchmarking protocols (Li et al., 2024a), we use an LLM judge (GPT-4o) (Zheng et al., 2023) to evalu-

ate whether a predicted answer matches the ground truth, and Contain-Match Accuracy (Contain-Acc.), which measures whether the response contains the correct answer. (2) Efficiency: We measure end-to-end efficiency from index construction to online inference, reporting token consumption and average wall-clock latency. (3) Implementations: Following HippoRAG2, we use Llama-3.3-70B-Instruct as the backbone LLM. All methods use the same embedding model all-mpnet-base-v2, and retrieve top-5 candidate passages/nodes per query. Since the answers of Medical are descriptive statements, we replace Contain-Acc with token-level F1. Details are provided in Appendix C.

## 4.2. Main Results

**Inference Performance Comparison.** Table 1 summarizes inference accuracy across datasets and task settings. (1) Overall gains on multi-hop and domain QA: GAM-RAG consistently outperforms all baseline families on multi-hop benchmarks and the domain-specific Medical dataset. The improvement is especially pronounced on MuSiQue, where GAM-RAG exceeds the second-best method by an average of 16.5% in GPT-Acc, highlighting the benefit of memory-driven hierarchical retrieval that enables more accurate targeting of evidence sentences throughout multi-hop traversal. (2) Time-sensitive QA: on TimeQA, DyG-RAG achieves the best performance, due to its event-graph construction for temporal anchoring; however, this design does not transfer as well to non-temporal tasks. In contrast, GAM-RAG achieves the second-best result in TimeQA, benefiting from its explicit time memory that helps to resolve semantically similar but temporally conflicting evidence. (3) Performance improves as memory stabilizes: we also observe a warm-up effect: in early episodes, retrieval can be slightly affected because only a small subset of sentence memories are activated

*Table 2.* **Efficiency and robustness analysis on 2Wiki.** We report offline indexing cost (token budget and wall-clock time) and online performance/latency under three query scenarios: Same Query, Similar Query, and Different Query, including the effect of multi-turn memorization for memory-based methods.

| Methods | Indexing | | Same Query | | Similar Query | | Different Query | |
|---|---|---|---|---|---|---|---|---|
| | Tokens(M) | Time(ks) | GPT-Acc | Time(s) | GPT-Acc | Time(s) | GPT-Acc | Time(s) |
| GFM-RAG | 4.16 | 3.57 | 59.80 | 4.16 | 57.40 | 4.07 | 56.30 | 4.23 |
| HippoRAG2 | 5.36 | 3.48 | 55.70 | 4.82 | 54.10 | 4.75 | 54.10 | 5.10 |
| LinearRAG | **0** | **0.54** | 61.70 | 1.32 | 60.30 | 1.45 | 60.60 | 1.52 |
| PoG | 4.97 | 3.57 | 54.10 | 20.43 | 51.90 | 20.10 | 51.10 | 21.32 |
| DyG-RAG | 9.54 | 7.75 | 40.70 | 9.86 | 40.20 | 8.45 | 39.80 | 9.71 |
| REMINDRAG | 4.28 | 3.21 | 62.10 | 13.25 | 61.60 | 17.62 | 60.70 | 20.86 |
| *-5-turn Memorization* | / | / | 65.90 | 7.92 | 65.30 | 7.43 | 63.50 | 8.33 |
| GAM-RAG | 1.66 | 1.03 | 65.30 | 7.43 | 64.60 | 7.82 | 62.90 | 10.98 |
| *-5-turn Memorization* | / | / | **71.80** | 3.11 | **71.00** | 3.27 | **69.80** | 3.61 |

and their uncertainties remain high; after more iterations, memories become more stable and reliable, translating into consistent performance gains.

**Efficiency and robustness analysis.** We further compare representative baselines on 2Wiki under the same evaluation protocol, with results reported in Table 2. (1) Efficiency. We first analyze the offline indexing cost (token budget and wall-clock time) of KG-based methods. Compared to approaches that require explicit relation extraction and KG materialization, GAM-RAG exhibits a clear advantage in index construction due to its lightweight hierarchical graph and simple sentence-level memory initialization. In contrast, LinearRAG incurs the smallest indexing overhead since it does not initialize sentence memories. More importantly, GAM-RAG reduces online latency through memory-guided traversal: similar to REMINDRAG, its memory mechanism helps localize evidence in fewer iterations by biasing propagation toward previously verified evidence. After 5 memorization turns on the same query, GAM-RAG achieves a 58% reduction in average retrieval time, suggesting that gain-adaptive memory enables reuse of prior evidence, thereby reducing redundant inference across repeated episodes.

(2) Robustness. Following the prior evaluation setup (Hu et al., 2025), we evaluate generalization under query perturbations by grouping historical queries into three scenarios: Same Query (identical to past queries), Similar Query (semantically equivalent paraphrases), and Different Query (different semantics but similar question types); details are provided in Appendix D. Methods without memory exhibit largely fixed performance on new queries, as they cannot reuse past outcomes. In contrast, GAM-RAG explicitly stores sentence-level memory and adapts its confidence via uncertainty-controlled gains, enabling dynamic activation of useful sentences over repeated queries. After 5 memorization turns, GAM-RAG improves average GPT-Acc by

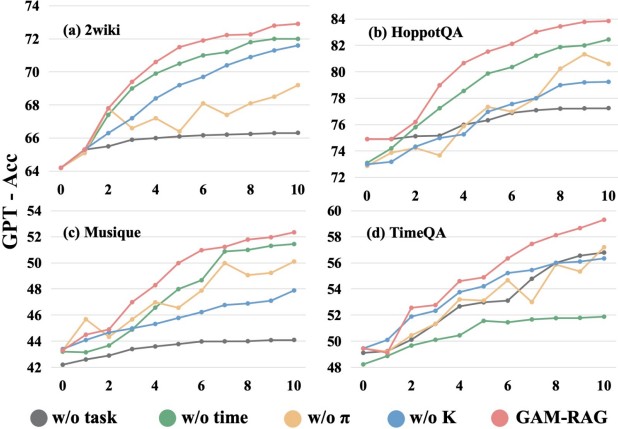

*Figure 4.* **Ablation under long-term memorization.** Performance trends over 10 turns on five datasets for GAM-RAG.

10.3% across the three settings and shortens query time by 61% *on average*, demonstrating that uncertainty-aware memory updates can simultaneously enhance robustness and efficiency in practice.

### 4.3. Ablation Study

To examine the long-term stability of GAM-RAG's sentence memory, we conduct a systematic ablation study. Specifically, we focus on four variants: (i) **w/o Task:** removes the task-memory vector $\mathbf{m}^{\text{task}}$ from both propagation and updating; (ii) **w/o Time:** removes the time-memory vector $\mathbf{m}^{\text{time}}$ from both propagation and updating; (iii) **w/o Uncertainty** $\pi$**:** removes confidence-aware scaling in propagation; (iv) **w/o Gain update** $K$**:** removes gain scheduling in memory updates by fixing the learning rate.

As shown in Fig. 4, we further track the ablation variants under a *10-turn* long-term memorization, and report how removing each module affects performance across datasets.

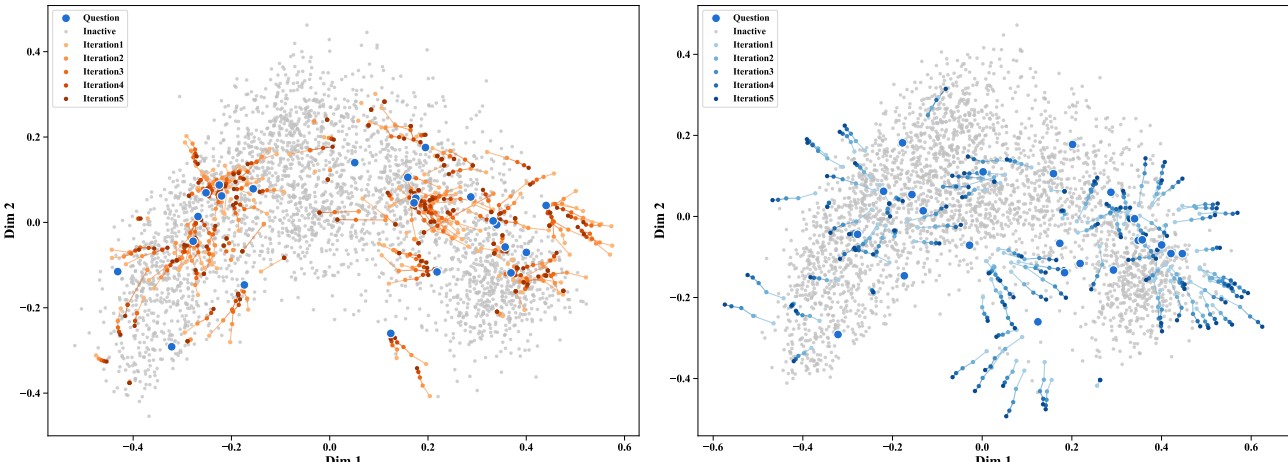

*Figure 5.* **Memory trajectories for supportive and non-supportive sentences.** We separately visualize the memory trajectories of supportive sentences and non-supportive sentences after iterative memory updates. Blue points denote queries, gray points denote inactive sentences, and colored points denote activated sentences, with lighter-to-darker colors indicating earlier-to-later memorization turns. The displacement of each activated sentence reflects the magnitude and direction of its memory update.

**(1) Long-term stability of GAM-RAG.** For the full model, accuracy exhibits a consistent upward trend over turns on all datasets, matching the expected fast warm-up followed by damped refinement behavior: early turns benefit from large gains and rapid correction of uncertain memories, while later turns converge as uncertainties decrease and updates become conservative. **(2) Task vs. time memories.** w/o $m_{\text{task}}$ causes substantial degradation on four datasets, as retrieval becomes primarily modulated by time memory. This limitation is especially pronounced on non-time-sensitive benchmarks, where time signals are weak and thus provide little additional discrimination. In contrast, on the time-sensitive benchmark, w/o $m_{\text{time}}$ leads to a clear drop. **(3) Uncertainty and gain scheduling**. The uncertainty controls how strongly memories influence propagation. When removing $\pi$, retrieval becomes more vulnerable to low-confidence memories, resulting in noticeable turn-to-turn fluctuations over long-term updates. Similarly, the $K$ governs the effective learning rate of memory adaptation. Replacing it with a fixed, small learning rate slows down memory correction, delaying performance improvements across turns.

### 4.4. Case Study of Memory

We conduct a case study on a subset of 2Wiki queries to visualize how sentence memories evolve during iterative updates. Specifically, we project sentence-memory vectors into a 2D space using PCA (Shlens, 2014) and track their trajectories over five memorization turns, as shown in Fig. 5. To make the update behavior clearer, we separately visualize supportive and non-supportive sentences. The results show that GAM-RAG learns different update directions for different types of evidence. Supportive sentences are gradually pulled toward the corresponding query

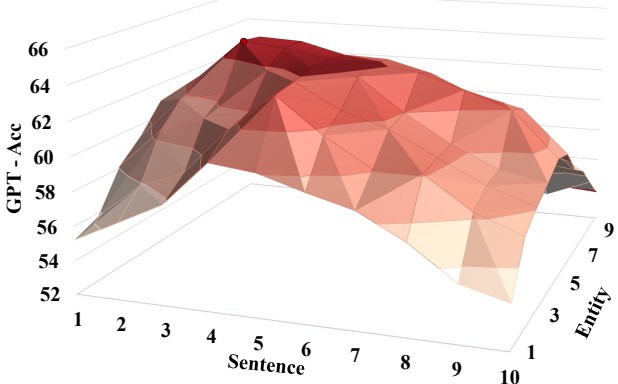

*Figure 6.* Search space analysis of GAM-RAG by varying the per-iteration activation for sentences ($K_S$) and entities ($K_E$).

regions, which strengthens their association with relevant query classes and makes them easier to retrieve in later interactions. In contrast, non-supportive sentences are pushed away from the query regions, reducing their influence in subsequent retrieval even when they are semantically close to the query. These trajectories provide an intuitive explanation of the adaptive memory mechanism. Instead of uniformly reinforcing all activated sentences, GAM-RAG selectively strengthens evidence that is verified as useful while weakening misleading or irrelevant evidence. This selective update process helps the memory form more stable query-evidence associations over repeated interactions.

### 4.5. Search Space Analysis

GAM-RAG constrains the exploration space by limiting the number of activated sentences and entities at each iteration.

*Table 3.* GPT-Acc (%) under different noisy-feedback ratios across memory update rounds.

| Noise Ratio | R0 | R1 | R2 | R3 | R4 | R5 | R6 | R7 | R8 | R9 | R10 |
|---|---|---|---|---|---|---|---|---|---|---|---|
| 0% | 64.2 | 65.3 | 67.8 | 69.4 | 70.6 | 71.5 | 71.9 | 72.2 | 72.3 | 72.8 | 72.9 |
| 10% | 63.8 | 64.6 | 64.4 | 66.5 | 66.2 | 69.3 | 70.5 | 71.2 | 71.7 | 72.1 | 72.4 |
| 20% | 61.8 | 62.6 | 62.0 | 63.9 | 63.4 | 65.7 | 68.0 | 69.0 | 69.8 | 70.6 | 71.8 |

In this section, we dynamically vary the maximum activated sentences $K_S$ and the activated entities $K_E$ per iteration. As shown in Fig. 6, moderate expansion improves evidence coverage, while overly large expansion introduces excessive noise and degrades retrieval quality. When $K_S$ is too small, only a few sentences participate in passage scoring, making the ranking overly sensitive to individual sentence weights and increasing the risk of missing complementary evidence needed for multi-hop reasoning. And when $K_S$ becomes large, many low-quality or weakly related sentences are activated, leading to performance degradation. While a small $K_E$ restricts graph expansion, which may fail to surface intermediate bridge entities that are necessary to connect distant hops. Beyond a moderate range, activating too many entities expands into semantically adjacent but irrelevant subgraphs, amplifying noise during propagation.

### 4.6. Long-term stability under noisy feedback.

We further examine whether the adaptive memory in GAM-RAG remains stable over long-term updates when noisy feedback exists. Since memory updates are confidence-aware, supportive evidence is reinforced only when it contributes to a correct answer, while unreliable feedback is assigned limited influence. This design helps prevent occasional noisy signals from being directly solidified into long-term memory. To evaluate this, we randomly inject feedback noise during memory updates with noise ratios of 0%, 10%, and 20%. Table 3 reports GPT-Acc across different update rounds. The results show that noisy feedback mainly causes early-stage fluctuations, while performance gradually stabilizes as memory confidence increases. Even under 20% noise, GAM-RAG continues to improve across update rounds rather than drifting or collapsing. This suggests that the adaptive memory mechanism remains stable under moderate feedback noise.

## 5. Conclusion

We present GAM-RAG, an experience-driven RAG framework that builds a relation-free hierarchical index and updates sentence-level task and time memories online. A Kalman-inspired, uncertainty-aware gain modulates these updates to adapt quickly while remaining robust to noisy feedback. On multi-hop and time-sensitive QA, GAM-RAG improves accuracy and lowers inference cost as experience accumulates, supporting training-free evolutionary retrieval.

## Acknowledgements

This work was supported by grants from the National Natural Science Foundation of China (72595845, 72595840).

## Impact Statement

This paper presents work on Retrieval-Augmented Generation (RAG) to advance dynamic memory mechanisms for large language models, which may offer useful perspectives for future research at the intersection of memory-augmented modeling and cognitive neuroscience. While the proposed framework may be subject to societal considerations similar to those of existing RAG systems, we do not identify any ethical concerns requiring specific emphasis beyond those generally associated with large language models and retrieval-based methods.

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

# A. Theoretical Analysis

## A.1. A Sufficient Condition for Memory Attraction/Repulsion

We provide a simple guarantee showing that, with an appropriate observation-noise level $R$, the gain-adaptive update can (i) *attract* a sentence memory toward a class of semantically similar queries when it is repeatedly judged as supportive, and (ii) *repel* it when it is repeatedly judged as non-supportive.

**Query class.** Fix a unit reference direction $\mathbf{q}^{(\cdot)}$ (task or time channel). For a threshold $\lambda \in (0, 1)$, define the query class as a spherical cap:

$$\mathcal{Q}_\lambda(\mathbf{q}^{(\cdot)}) = \left\{ \mathbf{q}' \in \mathbb{R}^d : \|\mathbf{q}'\|_2 = 1, \cos(\mathbf{q}', \mathbf{q}^{(\cdot)}) \geq \lambda \right\}. \tag{14}$$

**Assumption 1 (Consistent feedback on a query class).** Consider a fixed sentence $s_i$ and one channel $(\cdot) \in \{\text{task}, \text{time}\}$. Across $n$ consecutive retrieval episodes whose queries belong to $\mathcal{Q}_\lambda(\mathbf{q}^{(\cdot)})$, the judge yields a consistent signed target $y \in \{0, 1\}$ for $s_i$ (always supportive or always non-supportive).

**Proposition 1 (1D projected-state recursion).** Let

$$s_t \triangleq \cos(\mathbf{q}^{(\cdot)}, \mathbf{m}_{i,t}^{(\cdot)}) \in [-1, 1] \tag{15}$$

denote the projected support at episode $t$ (with $\|\mathbf{q}^{(\cdot)}\|_2 = 1$ and $\|\mathbf{m}_{i,t}^{(\cdot)}\|_2 = 1$ maintained by the post-update normalization). Then the update in Eq. 12 induces the 1D recursion

$$s_{t+1} = (1 - K_t)s_t + K_t y, \qquad \text{equivalently} \qquad |s_{t+1} - y| = (1 - K_t)|s_t - y|. \tag{16}$$

In particular, $s_t$ moves monotonically toward $y$: if $y = +1$ then $s_{t+1} \geq s_t$, and if $y = -1$ then $s_{t+1} \leq s_t$.

**Proof.** By definition, $s_t = \cos(\mathbf{q}^{(\cdot)}, \mathbf{m}_{i,t}^{(\cdot)})$ and the residual is $e_t = y - s_t$ (Eq. 10). Substituting into Eq. 12 gives

$$\mathbf{m}_{i,t+1}^{(\cdot)} \leftarrow \mathbf{m}_{i,t}^{(\cdot)} + K_t (y - s_t) \mathbf{q}^{(\cdot)}. \tag{17}$$

Taking inner product with $\mathbf{q}^{(\cdot)}$ (unit norm) and using $\langle \mathbf{q}^{(\cdot)}, \mathbf{m}_{i,t}^{(\cdot)} \rangle = s_t$ yields

$$s_{t+1} = s_t + K_t(y - s_t) = (1 - K_t)s_t + K_t y, \tag{18}$$

which proves the first equality in Eq. 16. Subtracting $y$ from both sides gives $s_{t+1} - y = (1 - K_t)(s_t - y)$, hence $|s_{t+1} - y| = (1 - K_t)|s_t - y|$. Monotonicity follows since $K_t \in (0, 1)$ implies $s_{t+1}$ is a convex combination of $s_t$ and $y$. $\square$

**Corollary 1 (Choosing $R$ to guarantee attraction/repulsion after $n$ episodes).** Because the process-noise offset satisfies $Q^{(\cdot)} > 0$ in Eq. 13, we always have $\pi_{i,t}^{(\cdot)} \in [Q^{(\cdot)}, 1]$, and therefore the gain admits a uniform lower bound:

$$K_t = \frac{\pi_{i,t}^{(\cdot)}}{\pi_{i,t}^{(\cdot)} + R} \geq \kappa(R) \triangleq \frac{Q^{(\cdot)}}{Q^{(\cdot)} + R}. \tag{19}$$

Unrolling Eq. 16 over $n$ consecutive episodes gives

$$|s_{t+n} - y| = \left( \prod_{j=0}^{n-1}(1 - K_{t+j}) \right) |s_t - y| \leq (1 - \kappa(R))^n |s_t - y|. \tag{20}$$

Hence, for any margin $\lambda \in (0, 1)$, there exists a choice of $R$ (equivalently $\kappa(R)$) such that the memory becomes *attracted* to, or *repelled* from, the query-class direction after $n$ consistent episodes:

$$y = +1 : \quad s_{t+n} \geq \lambda \quad \text{whenever} \quad (1 - \kappa(R))^n \leq \frac{1 - \lambda}{1 - s_t}, \tag{21}$$

$$y = -1 : \quad s_{t+n} \leq -\lambda \quad \text{whenever} \quad (1 - \kappa(R))^n \leq \frac{1 - \lambda}{1 + s_t}. \tag{22}$$

*Table 4.* Dataset summary and evaluation splits used in our experiments.

| Dataset | Setting | Corpus | Eval. questions |
|---|---|---|---|
| HotpotQA (Yang et al., 2018) | Multi-hop QA | Same as HippoRAG | 1,000 (val) |
| 2Wiki (Ho et al., 2020) | Multi-hop QA | Same as HippoRAG | 1,000 (val) |
| MuSiQue (Trivedi et al., 2022) | Multi-hop QA | Same as HippoRAG | 1,000 (val) |
| TimeQA (Chen et al., 2021) | Time-sensitive QA | Provided corpus | 3,078 (test-hard) |
| Medical (Xiao et al., 2025) | Domain QA | GraphRAG-Bench | 4,076 (all) |

Eq. 21 formalizes *attraction*: repeated positive feedback drives the projected support toward $+1$, making the memory increasingly aligned with queries in $\mathcal{Q}_\lambda(\mathbf{q}^{(\cdot)})$. Eq. 22 formalizes *repulsion*: repeated negative feedback drives the projected support toward $-1$, suppressing the memory's influence for that query class. In GAM-RAG, we implement asymmetric observation noise by setting $R = R_{\text{pos}}$ for $y = +1$ and $R = R_{\text{neg}}$ for $y = -1$ with $R_{\text{neg}} > R_{\text{pos}}$, which intentionally slows down repulsion to avoid prematurely discarding potential bridge evidence.

## B. Dataset details.

We briefly summarize the characteristics of each dataset below.

(i) **HotpotQA** (Yang et al., 2018). A classic multi-hop benchmark where answering typically requires aggregating evidence across multiple Wikipedia articles. Each question is paired with supporting passages but also includes substantial distractors, making evidence selection and cross-document reasoning essential.

(ii) **2WikiMultiHopQA (2Wiki)** (Ho et al., 2020). A large-scale multi-hop QA dataset constructed from Wikipedia, where instances require composing evidence from a fixed set of articles (commonly 2 or 4). It stresses structured reasoning across documents and maintaining coherent information flow over multi-step inference.

(iii) **MuSiQue** (Trivedi et al., 2022). A compositional multi-hop benchmark designed around sequential reasoning, often involving 2–4 dependent steps. Its questions emphasize consistent multi-step logical inference across multiple documents, rewarding retrieval that preserves contextual consistency throughout the reasoning chain.

(iv) **TimeQA** (Chen et al., 2021). A time-sensitive QA benchmark with implicit temporal expressions and cross-sentence temporal dependencies. We use the test-hard split (3,078 questions), which is particularly challenging due to temporal conflicts and requires robust time-aware evidence selection.

(v) **Medical (GraphRAG-Bench)** (Xiao et al., 2025). A domain-specific QA dataset derived from GraphRAG-Bench, built upon structured clinical knowledge sources such as guideline-style medical documents. It covers multiple task types with increasing difficulty (e.g., fact retrieval, complex reasoning, and contextual summarization), serving as a stress test for specialized-domain generalization.

## C. Implementation details.

- **Search Depth** ($I$). This parameter controls the maximum number of iterative expansion rounds in memory/graph search. A larger $I$ improves coverage for multi-hop evidence but can increase runtime and amplify spurious expansions, while too small $I$ may truncate valid reasoning chains. In practice, we find that $I \in \{1, 2, 3\}$ already covers most useful evidence for standard multi-hop QA benchmarks under a fixed context budget. We set $I = 3$ to balance recall and computational stability.

- **Memory Update Noise** ($R_{\text{pos}}$, $R_{\text{neg}}$). These parameters determine how aggressively the Kalman-style update reinforces or suppresses sentence memories based on feedback signals. Overly small noise makes updates too sharp and risks overfitting to occasional judge/retrieval errors (catastrophic forgetting), whereas overly large noise weakens memory adaptation and reduces the benefit of iterative refinement. We adopt an asymmetric design with $R_{\text{pos}} = 0.5$ and $R_{\text{neg}} = 1.0$. This setting also ensures sufficient memory coverage within our 10-round evaluation: with $R = 1.0$ and $R = 0.5$, the memory activation ratio after 10 updates reaches 0.90 and 0.95.

- **Per-hop Expansion Breadth.** We expand each hop with the top-3 sentences and use the top-5 next-hop entities as seeds for the following round. Increasing these values can improve exploration but quickly enlarges the frontier and

*Table 5.* Effect of different sentence embedding models on GAM-RAG, evaluated by GPT-Acc across five QA benchmarks.

| Methods | 2Wiki | HotpotQA | MuSiQue | TimeQA | Medical |
| --- | --- | --- | --- | --- | --- |
| | GPT-Acc | GPT-Acc | GPT-Acc | GPT-Acc | GPT-Acc |
| GritLM-7B (Muennighoff et al., 2024) | 64.00 | **75.10** | 43.10 | 49.11 | 65.24 |
| NV-Embed-v2 (Lee et al., 2024) | 64.10 | 74.60 | 43.20 | 48.75 | 64.98 |
| all-mpnet-base-v2 (Song et al., 2020) | 64.20 | 74.90 | **43.30** | **49.45** | **65.47** |
| all-MiniLM-L6-v2 (Wang et al., 2020) | 63.70 | 73.70 | 42.80 | 48.11 | 65.22 |
| bge-large-en-v1.5 (Xiao et al., 2024) | 64.10 | 74.30 | 43.10 | 48.73 | 64.28 |
| e5-large-v2 (Wang et al., 2024) | **65.40** | 73.80 | 42.90 | 49.23 | 65.42 |

raises the risk of hub domination; reducing them improves efficiency but may miss critical bridging evidence. Our setting keeps runtime stable across datasets while maintaining adequate multi-hop coverage within $I \leq 3$ rounds.

- **Passage-level Scoring Fusion.** We incorporate a weak dense-retrieval prior (ratio = 0.01). The small coefficients ensure dense similarity mainly serves as a tie-breaker instead of overwhelming evidence aggregated from activated sentences, preventing the method from degenerating into a pure dense retriever.

- **Initialization of Memory Uncertainty.** We initialize time/task uncertainty with $p_0 = 1.0$ to keep early gating neutral so that the system initially relies on semantic similarity; as updates accumulate, uncertainty decreases and memory signals become progressively more influential.

## D. Robustness Setup: Similar vs. Different Queries

**Similar Query (Paraphrased Equivalents).**    Following the *similar question rewriting* protocol adopted in prior work (Hu et al., 2025), we construct Similar Query instances by rewriting each original question to maximize surface-form differences while preserving its underlying intent and answer. Concretely, we use an LLM-based rewriting prompt (see Section F in Appendix) to generate paraphrases that keep the same semantics but vary lexical choices and phrasing.

**Different Query (Same Type, Different Semantics).**    Instead of constructing minimally perturbed questions with flipped answers as in prior work, we define Different Query by leveraging the $question_type$ annotation in the *2Wiki* dataset. We partition the dataset into disjoint subsets by question type (e.g., comparison, compositional), and further split each subset into two equal parts. Models are exposed to one split and evaluated on the other, ensuring that test queries share the same task format but differ in semantics and answers. This setting probes whether a method generalizes across unseen question content within the same reasoning type, rather than memorizing specific historical queries.

## E. Additional Experiments

### E.1. Effect of sentence embedding models.

We study how the choice of sentence embedding model affects retrieval performance by swapping the encoder while keeping the retriever pipeline and all other settings fixed. Across five benchmarks, as shown in Table 5, we observe that the overall performance differences among the evaluated embedding models remain relatively small, and the ranking is broadly consistent across datasets. This suggests that the dense-retrieval backbone is fairly robust to the specific embedding choice under our evaluation protocol. Based on this observation and its favorable efficiency–performance trade-off, we adopt all-mpnet-base-v2 as the default embedding model in subsequent experiments.

### E.2. Robustness to query ordering.

Since GAM-RAG updates memory states by accumulating experience from previous queries, a natural concern is whether its performance depends heavily on a particular query order or repeated query patterns. To address this concern, we further evaluate the robustness of GAM-RAG under shuffled query sequences. Specifically, based on the same settings used in the main experiments, namely Same Query, Similar Query, and Different Query, we randomly shuffle the query order and conduct 10 independent runs for each method. We report the mean GPT-Acc and the corresponding standard deviation in Table 6.

*Table 6.* GPT-Acc (%) of different methods over 10 shuffled query runs.

| Methods | Same Query | Similar Query | Different Query |
|---|---|---|---|
| HippoRAG2 | $55.68 \pm 0.06$ | $54.08 \pm 0.07$ | $54.06 \pm 0.07$ |
| LinearRAG | $61.68 \pm 0.03$ | $60.28 \pm 0.05$ | $60.56 \pm 0.06$ |
| REMINDRAG | $62.03 \pm 0.18$ | $61.47 \pm 0.21$ | $60.52 \pm 0.25$ |
| GAM-RAG | $65.26 \pm 0.15$ | $64.50 \pm 0.18$ | $62.76 \pm 0.22$ |
| GAM-RAG (-5-turn Memorization) | $\mathbf{71.58 \pm 0.37}$ | $\mathbf{70.82 \pm 0.42}$ | $\mathbf{69.41 \pm 0.50}$ |

*Table 7.* EM / F1 (%) results across QA benchmarks.

| Methods | 2Wiki | HotpotQA | MuSiQue | TimeQA |
|---|---|---|---|---|
| GFM-RAG | 58.5/64.8 | 64.2/61.0 | 33.0/28.0 | 41.5/44.0 |
| HippoRAG2 | 54.3/61.2 | 62.8/62.1 | 23.5/26.6 | 40.8/43.2 |
| LinearRAG | 60.2/68.5 | 64.8/62.6 | 35.8/31.7 | 43.5/47.0 |
| PoG | 52.6/60.0 | 56.5/60.2 | 30.1/28.5 | 40.6/42.4 |
| REMINDRAG | 59.8/67.5 | 72.6/70.5 | 33.8/30.2 | 42.5/45.0 |
| GAM-RAG | $\mathbf{62.8/71.0}$ | $\mathbf{73.5/71.8}$ | $\mathbf{40.5/38.2}$ | $\mathbf{47.8/50.5}$ |

The results show that the variation across different query orders is consistently small for all methods. More importantly, GAM-RAG maintains clear performance advantages over the compared baselines under all three query settings. This indicates that the gains of GAM-RAG are not driven by a specific query sequence. Instead, the memory update mechanism remains stable when the order of incoming queries changes, suggesting that GAM-RAG is robust to query shuffling and does not rely on a fixed repeated-query pattern.

### E.3. Automatic evaluation metrics.

To facilitate comparison with prior QA studies, we further report automatic evaluation results using Exact Match (EM) and token-level F1 on four benchmarks. The results are shown in Table 7. GAM-RAG consistently outperforms the compared baselines under both EM and F1, which is consistent with the trends observed under GPT-based and Contain-Acc evaluation.

## F. Prompts

Here are the prompt templates we used, including those for time extraction, sufficiency verification, obtaining the sentence observation signal, and final answer generation.

> **Time Extraction Prompt Template**
>
> You are extracting time constraints for reasoning. Given a sentence, extract ONLY the time-related constraint that limits when the statement holds.
>
> Rules:
> - Output a short phrase, not a full sentence.
> - Extract explicit, implicit, or relative time constraints if present.
> - If no time constraint is present, output: NONE.
>
> Sentence:
> "SENTENCE"
> Time constraint:

**Sufficiency Verification Prompt Template**

You are a evidence sufficiency judge. You will be given a Question and N Sentences. Your task: determine whether the sentences contain enough information to answer the question.

Rules:
1) If the sentences contain sufficient evidence to answer confidently, output YES.
2) If the sentences are missing key facts needed to answer, output NO.
3) Output STRICT JSON only, with no extra text.

Return EXACTLY this schema:
"sufficient": "yes" or "no"

**Sentence Observation Signal Prompt Template**

You will receive: a Question, a Predicted Answer, and a list of Sentences, each with a sid.

Task: Return the sids of sentences that are SUPPORT.
What counts as SUPPORT
- Direct evidence that states the answer, OR
- Bridge / clue for multi-hop reasoning: a necessary intermediate fact, constraint, definition, relation
- Output STRICT JSON only.

Return:
"support sids": ["sid", ...]

**Final Answer Generation Prompt Template**

As an advanced reading comprehension assistant, your task is to analyze text passages and corresponding questions meticulously. Your response start after "Thought: ", where you will methodically break down the reasoning process, illustrating how you arrive at conclusions. Conclude with "Answer: " to present a concise, definitive response, devoid of additional elaborations.

**The Similar Questions Rewriting Prompt Template**

Now I'll give you a question, and I want you to rewrite it as a different question that means the same thing but is phrased differently. Original question: query When you respond, just give me your rewritten question.

