# OpenReview forum: "GAM-RAG: Gain-Adaptive Memory for Evolving Retrieval in Retrieval-Augmented Generation"
_ICML.cc/2026/Conference — ICML 2026 regular_

### Official Review · Reviewer_HZnP · 2026-03-07

**Soundness:** 2
**Presentation:** 2
**Significance:** 2
**Originality:** 2
**Overall Recommendation:** 4
**Confidence:** 2

**Summary:**

The manuscript's key topic consists of a training-free framework called GAM-RAG, designed to evolve retrieval structures through inference experience. The authors attempt to address an important context where traditional RAG systems remain static and redundant when handling recurring or multi-hop queries. By introducing a Kalman-inspired update rule and a relation-free hierarchical index, the paper aims to balance memory plasticity with stability against noisy feedback.

**Compliance With Llm Reviewing Policy:**

Affirmed.

**Final Justification:**

My concern has been addressed.

**Key Questions For Authors:**

How does this method compare with other agent memory works, such as Amem and Mem0?

**Limitations:**

Limited evaluation on agent memory benchmarks.

**Strengths And Weaknesses:**

Strengths:

1. The application of "Schema-based learning" from cognitive neuroscience to RAG is refreshing and logically sound for solving the "stateless" limitation of current systems.

2. A 61% reduction in inference cost is highly significant.

3. The use of Kalman Gain to modulate updates based on Perplexity (Uncertainty) is a mathematically grounded way to handle the inherent noise in LLM retrieval.

Weakness:
1. The evaluations are mainly based on a multi-hop QA dataset, such as 2wiki. However, I believe the comparison with other baselines, such as the dataset as Locomo and methods like Amem, mem0, is necessary.

2. While the Kalman-inspired update rule aims to balance stability and plasticity, the system may still be susceptible to memory drift. If the initial retrieval episodes contain latent biases or if the model is exposed to a series of semantically narrow queries, the memory states might overfit to a specific "reasoning sub-domain." The manuscript lacks a discussion on a forgetting mechanism or a "reset" protocol for stale or corrupted memories that might misguide future retrieval。

3. The core of the Gain-Adaptive mechanism relies on LLM Perplexity as a proxy for uncertainty. However:

Model Dependency: Perplexity distributions vary significantly across different LLM architectures (e.g., Llama vs. Mistral). The robustness of the "Gain" across different generator models is not fully explored.

Calibration: High perplexity does not always equate to "noisy retrieval"; it can also stem from inherently complex linguistic structures in the ground truth. This ambiguity might lead to the improper damping of valid, high-quality update

---

> ### Author Rebuttal · Authors · 2026-03-30
>
> We sincerely thank you for this insightful question and we clarify the key points below.
>
> > ### **Regarding W1 and Q1 : Compared to Agentic Memory Methods**
>
> We respectfully note that these two lines of work address **different memory problems** and are designed for **different application settings**, so a direct empirical comparison may not be fully appropriate.
>
> 1. **Retrieval optimization vs. long-context memory.**
>
>    GAM-RAG studies how to better **retrieve** useful evidence from an existing external knowledge base by updating retrieval preferences. In contrast, methods such as **Mem0** focus on how to **store** useful information from long conversations so that it can influence later response generation.
>
> 2. **Different roles of memory.**
>    In GAM-RAG, memory operates on top of a pre-built external knowledge index and affects **retrieval preference** during search. In agent memory methods, memory is typically stored as text snippets from interaction history and later retrieved mainly through simple similarity matching to support generation.
>
> 3. **Different evaluation settings.**
>
>    GAM-RAG is evaluated on benchmarks such as **2Wiki**, where the goal is to measure retrieval quality over an existing knowledge base. Agent memory methods are typically evaluated on datasets such as **LoCoMo**, where the goal is to measure how well user-specific information from long dialogues is stored and reused.
>
> For these reasons, we believe these methods are best viewed as **complementary rather than directly competing**. To further address this concern, we additionally evaluate A-Mem and Mem0, on the LoCoMo dataset under two retrieval settings: their original top-$k$ retrieval and retrieval replaced by **GAM-RAG**.
>
> Table 1:  F1 / BLEU-1 results on LoCoMo
>
> | Method          | Single Hop (F1 / B1) | Multi-Hop (F1 / B1) | Open Domain (F1 / B1) | Temporal (F1 / B1) |
> | --------------- | -------------------: | ------------------: | --------------------: | -----------------: |
> | A-Mem           |          28.5 / 21.3 |         13.4 / 12.5 |           45.6 / 37.9 |        46.9 / 37.5 |
> | A-Mem + GAM-RAG |          30.1 / 22.6 |         16.0 / 14.5 |           48.1 / 39.8 |        49.8 / 39.9 |
> | Mem0            |          40.2 / 28.4 |         29.5 / 22.4 |           49.0 / 39.6 |        50.6 / 41.4 |
> | Mem0 + GAM-RAG  |      **41.7 / 29.5** |     **32.8 / 24.9** |       **51.4 / 41.3** |    **53.4 / 43.3** |
>
> Table 1 shows that GAM-RAG consistently improves retrieval quality when integrated into agentic memory methods.
>
> > ### **Regarding W2 : Mitigating Memory Drift**
>
> We clarify that GAM-RAG mitigates memory drift in practice through two components: memory bias mitigation during retrieval and update, and a memory reset rule.
>
> * First, passage retrieval is not determined by a single memory state. The final passage score is aggregated from multiple activated sentence scores, so the bias of any single sentence memory has limited influence. (Eq. (7))
>
> * Second, in our setup, confidence is increased only for supportive sentences, which already makes memory reinforcement selective, and the offset term $Q$ helps prevent confidence collapse and reduces premature fixation. (Eq. (13))
>
> * Finally, we use a simple reset rule: if a sentence is retrieved **three consecutive times** and judged as **non-supportive**, its memory is reset to the original sentence semantic representation.
>
> This is also supported by our long-term stability analysis under noisy feedback, where performance shows only early-stage fluctuations and gradually stabilizes as memory confidence accumulates (see Table 2 in our response on Long-Term Stability Analysis under noisy feedback).
>
> We have added a clearer discussion of this mechanism in the Implementation Details.
>
> > ### **Regarding W3: Clarification of Memory Perplexity and Gain Computation**
>
> We would like to gently clarify a possible misunderstanding of our definition. We respectfully refer to Sec. 3.3.1 of the manuscript. The memory **perplexity** $\pi_i$ in our method is **not** the LLM perplexity of the generator. As noted above, this memory perplexity is updated only when the sentence is judged as supportive. In the revised manuscript, we have replaced this expression with **uncertainty** to better reflect its role in our method.
>
> In our framework, the LLM judge only provides a binary support signal $y_i \in \{0,1\}$, which is used to compute the residual. The Kalman gain is then computed as
> $$
> K_i^{(\cdot)}=\frac{\pi_i^{(\cdot)}}{\pi_i^{(\cdot)}+R_i^{(\cdot)}},
> $$
> where $R_i$ is a hyperparameter controlling the update step size, and $\pi_i$ is a sentence-level memory uncertainty initialized to 1 and updated by Eq. (13) when the sentence is activated.
>
> To further address sensitivity to the LLM judge, we have also added an additional analysis with different judge backbones (see Table 2 in our response on Robustness of the LLM Judge). We hope this clarifies the definition and role of $\pi_i$.

---

> > ### Author Rebuttal · Reviewer_HZnP · 2026-04-01
> >
> > The rebuttal has addressed my concern. I will increase my score to 4

---

> > > ### Author Response · Authors · 2026-04-02
> > >
> > > Dear Reviewer HZnP,
> > >
> > > Thank you very much for your positive feedback and for your careful consideration of our rebuttal. We sincerely appreciate your recognition that your concerns have been adequately addressed. Your comments and suggestions have been very helpful in improving our manuscript.
> > >
> > > Best regards,
> > >
> > > Authors of Paper34058

---

### Official Review · Reviewer_ot7E · 2026-03-11

**Soundness:** 3
**Presentation:** 3
**Significance:** 3
**Originality:** 2
**Overall Recommendation:** 4
**Confidence:** 4

**Summary:**

This paper proposes a training-free retrieval-augmented generation (RAG) framework named GAM-RAG, aiming to address the issue that existing RAG systems tend to perform repeated multi-hop retrieval when processing relevant or similar queries due to their static retrieval indices, thereby increasing latency and computational costs.
Inspired by schema learning in cognitive neuroscience, this method constructs a lightweight relational hierarchy-free graph index to accumulate retrieval experience. Its main contribution lies in introducing a gain-adaptive updating mechanism inspired by Kalman filtering, which can dynamically and jointly update sentence-level memory states and their uncertainty (perplexity) based on retrieval feedback, thus achieving an optimal balance between rapidly absorbing reliable new signals and resisting noisy feedback.
Experimental results show that compared with the strongest baseline model, GAM-RAG improves the average performance by 3.95% (up to 8.19% when combined with 5-round memory), while substantially reducing inference costs by 61%.

**Compliance With Llm Reviewing Policy:**

Affirmed.

**Final Justification:**

I have read the other reviews and choose to accept this paper.
But I still have the concerns for the novelty of this paper.
I have raised the score to 4.

**Key Questions For Authors:**

See weakness.

**Limitations:**

yes

**Strengths And Weaknesses:**

Strengths:

* It accumulates retrieval experience during inference without retraining, effectively reducing redundant computations when processing similar queries.

* Adopting an adaptive mechanism inspired by Kalman filtering, it can quickly absorb reliable new knowledge while avoiding overfitting to erroneous feedback.

* Compared with the strongest baseline, it improves the accuracy by up to 8.19% and reduces the online inference cost by 61%.

Weaknesses:

* The framework needs to assign and maintain independent task memory and time memory vectors for every single sentence in the graph index. For enterprise knowledge bases with millions of documents, such extremely fine‑grained state storage incurs a catastrophic burden, raising serious concerns about scalability.

* Both during graph index construction and user query processing, the system relies entirely on lightweight tools (such as spaCy) for NER to perform node activation. Poor extraction performance leads to error accumulation, which further causes retrieval failures. NER requires further detailed evaluation and discussion.

---

> ### Author Rebuttal · Authors · 2026-03-30
>
> Dear Reviewer ot7E,
>
> Thank you for this insightful question and for giving us the opportunity to clarify these points. Below are detailed responses to your comments:
>
> > ### **Regarding W1 : Large-Scale Knowledge Base Deployment**
>
> We fully agree that storage and retrieval overhead are important considerations for any RAG system, and this was one of the design factors we explicitly considered in this work. Regarding Large-Scale Knowledge Base Deployment, we clarify that GAM-RAG does not require offline initialization of task/time memory for all sentences.
>
> Owing to the sparse hierarchical graph and retrieval design, the system does not compute similarity over all sentence vectors and select top-$k$ \. In each retrieval episode, only a small number of sentences activated during **Entity-Sentence propagation** (Sec. 3.2.2 in the manuscripts) participate in memory computation and update.
>
> Therefore, under large-scale deployment, sentence memory can be initialized **on demand** for the activated sentences and updated thereafter. We further compare this strategy with full pre-allocation in Table 1.
>
> Table 1:  trade-off between static pre-allocation and dynamic memory allocation. Storage overhead is normalized to the original GAM-RAG setting (100%).
>
> | Methods                                     | Storage Overhead (%) | Indexing Tokens (M) | Indexing Time (ks) | Query GPT-Acc | Query Time (s) |
> | ------------------------------------------- | -------------------: | ------------------: | -----------------: | ------------: | -------------: |
> | GAM-RAG                                     |                 100% |                1.66 |               1.03 |         65.30 |       **7.43** |
> | *-5-turn Memorization*                      |                 100% |                   / |                  / |         71.80 |       **3.11** |
> | GAM-RAG (Dynamic Allocation)                |              **26%** |               **0** |           **0.54** |         65.30 |           8.32 |
> | *-5-turn Memorization* (Dynamic Allocation) |              **32%** |                   / |                  / |         71.80 |           3.34 |
>
> As shown there, dynamic memory requires only one additional memory initialization and storage step for activated sentences during retrieval, shortens indexing time, and substantially reduces unnecessary vector storage. This allows users to flexibly choose the memory allocation strategy according to the scale of deployment.
>
> > ### **Regarding W2 : Robustness to NER Failures**
>
> 1. **When no entities can be extracted from the query**
>
>    We respectfully refer to Sec. 3.2.2 (Entity-Sentence propagation). When no entities can be extracted from the query, we skip the entity-to-sentence propagation step and directly compute sentence–query similarity $w_i^{\text{sent}}$ to initialize candidate sentences.
>    $$
>    w_i^{\text{sent}} = s_i^{\text{sem}} \cdot g_i^{\text{task}} \cdot g_i^{\text{time}}.
>    $$
>
>    This ensures robustness for entity-free or highly abstract queries.
>
> 2. **Entity ambiguity and synonymy.**
>
>    We recognized this as a potential false cause in graphRAG systems. Therefore, we have taken this into consideration when designing our retrieval algorithm. As described in Sec. 3.2.2, Even when ambiguous or noisy entities are activated, the framework relies on fine-grained semantic signals from sentence representations and memory alignment. As shown in Eq. (6):
>    $$
>    \text{score}_t(s_i) = \mathrm{Norm}_1\\left(w_i^{\text{sent}} \cdot u_t(s_i)\right).
>    $$
>    The final score combines semantic similarity and memory-guided weights, allowing correctly aligned sentences to dominate the propagation. This mitigates the effect of incorrect entity activation and promotes relevant intermediate nodes.
>
> 3. **Impact of different NER tools.**
>
>    We replace spaCy with two representative alternatives: a strong supervised baseline (RoBERTa-large NER) and an LLM-based extractor (GPT-4o). The results are summarized below:
>
>    Table 2: GPT-Acc (%) under different NER tools on 2Wiki.
>
>    | NER Method         | GPT-Acc (%) |
>    | ------------------ | ----------: |
>    | spaCy              |       64.20 |
>    | RoBERTa-large NER  |       64.40 |
>    | GPT-4o (LLM-based) |       64.70 |
>
> Overall, NER serves as a lightweight initialization signal, substantially reducing token consumption during the indexing stage, while the subsequent semantic and memory-based propagation ensures robustness to extraction errors.

---

> > ### Author Rebuttal · Reviewer_ot7E · 2026-04-01
> >
> > I will keep the score.

---

> > > ### Author Response · Authors · 2026-04-02
> > >
> > > Dear Reviewer ot7E,
> > >
> > > Thank you for carefully reading our previous response and for providing your valuable feedback. We sincerely appreciate your time and the opportunity to further improve our manuscript through this constructive exchange.
> > >
> > > **However, we regret that we are currently uncertain about the specific aspects of our manuscript that may still raise your concerns.**
> > >
> > > If possible, we would be very grateful if you could let us know, and we would be more than happy to provide further clarification or additional responses. Also, if appropriate in your final assessment, we would also be grateful if you might consider updating the score accordingly.
> > >
> > > Once again, thank you for your thoughtful feedback and guidance.
> > >
> > > Best regards,
> > >
> > > Authors of Paper34058

---

### Official Review · Reviewer_FJyF · 2026-03-11

**Soundness:** 3
**Presentation:** 3
**Significance:** 3
**Originality:** 3
**Overall Recommendation:** 4
**Confidence:** 3

**Summary:**

This paper proposes GAM-RAG, a training-free retrieval framework that aims to let a RAG system accumulate experience from past retrieval episodes without re-training the model. The proposed method organizes documents into a hierarchical architecture comprising entities, sentences, and passages. The retrieval workflow begins by identifying core entities and progressively activating a relational subgraph to discover relevant evidence. To ensure robustness against noisy feedback and maintain long-term stability, the authors introduce a Kalman-inspired gain update rule. This mechanism dynamically adjusts memory states by balancing memory uncertainty with sentence effective labels.

**Compliance With Llm Reviewing Policy:**

Affirmed.

**Final Justification:**

My concerns have been addressed, and I appreciate the authors' efforts. I am raising my score to 4 (Weak accept) and lean toward acceptance.

**Key Questions For Authors:**

1. The paper mentions that additional details are provided in the Appendix, but the submitted version does not appear to include an Appendix.

2. For QA benchmarks such as 2Wiki and HotpotQA, the community more commonly reports automatic metrics such as EM/F1. However, this paper instead uses GPT-4o as a judge and reports Contain-Acc. Contain-Acc may produce cases where the generated text contains the answer but the overall response is still incorrect. Moreover, relying on a single LLM as a judge may introduce bias and makes it difficult to directly compare with prior work. Could the authors also provide results using automatic metrics?

3. Could Table 5 be split into two figures, separating supportive and non-supportive sentences?

4. The paper would benefit from additional explanation and analysis of the process where an LLM judges the usefulness of each retrieved sentence.

**Limitations:**

yes

**Strengths And Weaknesses:**

Strengths

1. This paper attempts to enable RAG to accumulate experience during inference, gradually improving retrieval efficiency for future queries. This idea is meaningful for scenarios with many same or similar queries.

2. The hierarchical progressive exploration activation strategy works well. Even GAM-RAG without memory updates shows clear advantages. GAM-RAG with several rounds of memory updates further lead to clear performance gains

3. The introduction of a time attribute in both memory and uncertainty expands the applicability of the method, allowing it to be used for time-sensitive tasks.

Weakness

1. The paper refers to $\pi$ as perplexity, but it's not perplexity in the standard language modeling sense. While I understand that $\pi$ is intended to represent uncertainty, calling it perplexity may confuse readers.

2. In the main experiments, all GraphRAG-style methods use the same embedding model (all-mpnet-base-v2). However, Vanilla RAG is compared with three different embedding models for retrieval.

3. In Table 5, the authors use PCA to visualize the effect of memory updates. However, the figure mixes both supportive and non-supportive sentences. Apart from the labeled S1 and S2, it is unclear which sentences belong to which category. It would be clearer to separate supportive and non-supportive sentences into two figures to better illustrate how their positions relative to the query change.


4. Using an LLM to judge whether each retrieved sentence provides useful information introduces additional computational cost, but the paper does not analyze this overhead. Additionally, it is unclear whether this sentence-level support label is sensitive to the judge prompt or the judge model.

---

> ### Author Rebuttal · Authors · 2026-03-31
>
> Dear Reviewer FJyF，
>
> Thank you for your thoughtful and insightful feedback. Your comments have been helpful in improving the clarity of our work.
>
> > ### **Regarding W1 and Q1 : Terminology Clarification for $\pi$ and Appendix Availability**
>
> We agree that the term perplexity may be misleading here. In the revised manuscript, we have replaced this expression with **uncertainty** to better reflect its role in our method. We also clarify that the Appendix is provided in the **Supplementary Material**.
>
> > ### **Regarding W2 : Embedding Model Choice**
>
> We use the same embedding model for all GraphRAG-style methods to focus on the effect of the retrieval framework itself. For Vanilla RAG, we additionally report results with multiple embedding models because it serves as the most direct dense-retrieval baseline and this comparison helps show the sensitivity of standard dense retrieval to the encoder.
>
> To further address this point, we respectfully refer to Appendix E in the Supplementary Material, where we provide a dedicated analysis of different sentence embedding models. The results show that the overall differences across embedding models are relatively small and the ranking is broadly consistent across datasets.
>
> https://anonymous.4open.science/r/fig-table/fig1.png
>
> > ### **Regarding Q2 : Automatic Metrics**
>
> We used GPT-4o and Contain-Acc by following recent benchmarking protocols (LinearRAG), mainly because these datasets often admit semantically correct but non-identical answers, while the Medical benchmark contains many descriptive domain-specific responses for which exact-match-style metrics are less suitable.
>
> That said, we fully agree that automatic metrics improve comparability with prior work.  The results are presented below:
>
> Table 1: EM / F1 (%) across datasets.
>
> | Methods     |   2Wiki | HotpotQA | MuSiQue |  TimeQA |
> | ----------- | --------------: | ---------------: | --------------: | --------------: |
> | GFM-RAG     |     58.5 / 64.8 |      64.2 / 61.0 |     33.0 / 28.0 |     41.5 / 44.0 |
> | HippoRAG2   |     54.3 / 61.2 |      62.8 / 62.1 |     23.5 / 26.6 |     40.8 / 43.2 |
> | LinearRAG   |     60.2 / 68.5 |      64.8 / 62.6 |     35.8 / 31.7 |     43.5 / 47.0 |
> | PoG         |     52.6 / 60.0 |      56.5 / 60.2 |     30.1 / 28.5 |     40.6 / 42.4 |
> | REMINDRAG   |     59.8 / 67.5 |      72.6 / 70.5 |     33.8 / 30.2 |     42.5 / 45.0 |
> | **GAM-RAG** | **62.8 / 71.0** |  **73.5 / 71.8** | **40.5 / 38.2** | **47.8 / 50.5** |
>
> The results show that  GAM-RAG consistently outperforms the compared baselines, and we have incorporated these results into the updated version.
>
> > ### **Regarding W3 and Q3 : Clearer Memory Visualization**
>
> We thank the reviewer for this helpful suggestion. Accordingly, in the revised manuscript, we have placed supportive and non-supportive sentences into two separate figures, which more clearly shows how their positions relative to the query change after memory update, as shown below:
>
> https://anonymous.4open.science/r/fig-table/fig2.png
>
> > ### **Regarding W4 and Q4: Overhead and Robustness of the LLM Judge**
>
> - **LLM-Judge Overhead.**
>
>   We clarify that the LLM judge is not applied to each retrieved sentence individually. Instead, it is called once per retrieval iteration over the small set of sentences activated during Entity-Sentence propagation.
>
>   As described in Supplementary Material C , we use top-3 sentences per step and set the maximum search depth to 3, so at most **9 sentences** are passed to the judge in one query. The detailed prompt template as follows:
>
>    https://anonymous.4open.science/r/fig-table/fig3.png
>
>   In addition, we respectfully refer to Sec. 4.2. Unlike LLM-guided multi-step retrieval methods, GAM-RAG performs retrieval through semantic and memory-based propagation, and only applies a single judge call for memory update. As a result, it remains competitive in efficiency among the compared baselines.
>
> * **LLM-Judge Robustness.**
>
>   After retrieval, we collect the  activated sentences and apply a LLM judge to identify which ones are supportive. The prompt provides the question, predicted answer, and sentence list, defines support using only two criteria (**direct evidence** or a **necessary bridge clue**). In addition, the support label is only used as a lightweight observation signal for memory update, rather than as the sole retrieval score, which further limits its sensitivity.
>
>   We fully agree that the choice of judge model should be examined. To address this, we add an analysis with different backbone LLM judges:
>
>   Table 2: EM/F1 (%) on 2Wiki under different LLM judges.
>
>   | Backbone               |     EM / F1 |
>   | ---------------------- | ----------: |
>   | Llama-3.1-8B-Instruct  | 62.0 / 70.3 |
>   | Qwen3–32B              | 62.4 / 70.7 |
>   | Llama-3.3-70B-Instruct | 62.8 / 71.0 |
>   | GPT-4o                 | 62.8 / 71.2 |
>
> The results confirm that the overall trend remains stable across judge models.

---

> > ### Author Rebuttal · Reviewer_FJyF · 2026-04-02
> >
> > Thank you for the response. The rebuttal address my concerns and I will raise my score to 4.

---

> > > ### Author Response · Authors · 2026-04-02
> > >
> > > Dear Reviewer FJyF,
> > >
> > > Thank you very much for your positive feedback and for your careful consideration of our rebuttal. We sincerely appreciate your recognition and support.
> > >
> > > Best regards,
> > >
> > > Authors of Paper34058

---

### Official Review · Reviewer_rmfz · 2026-03-13

**Soundness:** 2
**Presentation:** 3
**Significance:** 2
**Originality:** 3
**Overall Recommendation:** 4
**Confidence:** 2

**Summary:**

This paper proposes GAM-RAG, a retrieval-augmented generation framework that introduces an adaptive memory mechanism on top of a graph-based retrieval structure. Instead of using a static retrieval index, the system maintains a memory state and uncertainty estimate for each node in the retrieval graph and updates them over time based on feedback from previous reasoning episodes. The goal is to allow the RAG system to accumulate experience across queries, improving retrieval quality and reducing search cost in multi-hop question answering tasks.

**Compliance With Llm Reviewing Policy:**

Affirmed.

**Key Questions For Authors:**

- How stable is the adaptive memory over long deployments? If the system processes a very large number of queries, does the retrieval graph converge to a stable state or continue drifting over time?

- Since the approach relies on accumulating experience from past queries, how much do the results depend on query ordering or repeated query patterns during evaluation? Have the authors tested robustness under shuffled or non-repetitive query sequences?

**Limitations:**

Yes

**Strengths And Weaknesses:**

**Strengths:**

- The paper explores an interesting direction of making RAG systems adaptive rather than static, which is a natural extension for long-running QA systems that may benefit from accumulating retrieval experience across queries.

- The approach is training-free and operates at inference time, which is appealing from a practical perspective since it avoids retraining large language models or retrievers.

- Experiments on multi-hop QA benchmarks show consistent improvements in answer accuracy and reductions in retrieval cost compared to static RAG baselines.

**Weaknesses:**

- The motivation for moving beyond static RAG could be clearer. Static retrieval systems have advantages such as stability, reproducibility, and simplicity, and it is not fully demonstrated that the benefits of adaptive memory outweigh these advantages in realistic settings.

- Because the method relies on sequential experience accumulation, the evaluation may be sensitive to query ordering or repeated query patterns. It is unclear whether the gains would remain when queries are shuffled or when the system encounters diverse queries without repetition.

- The paper does not thoroughly analyze long-term stability. Continuous updates to the retrieval graph could potentially lead to drift or reinforce incorrect evidence if the feedback signals are noisy.

---

> ### Author Rebuttal · Authors · 2026-03-30
>
> Dear Reviewer rmfz，
>
> Thank you for recognizing our work and for providing such thorough and insightful feedback. Below are detailed responses to your comments and suggestions:
>
> > ### **Regarding W1 : The Advantage of Adaptive Memory**
>
> We fully agree that the motivation for moving beyond static RAG should be clearer. To address this, we revised the motivation in the manuscript to make our intended direction more explicit. Specifically:
>
> - **Redundant Computation.**
>
>   Static RAG processes each query independently, so related or recurring queries often repeat similar traversal and reasoning steps.
>
> - **Limited Adaptability.**
>
>   Once the index is built, it remains fixed and cannot incorporate feedback from historical retrievals.
>
> To address this concern, we further clarify and validate these aspects as follows:
>
> - **Stability and Reproducibility.**
>
>   We evaluate robustness under different query types with **10 shuffled runs** (Table 1), and further analyze memory-state evolution under long-term deployment (Table 2).
>
> - **Simplicity and Efficiency.**
>
>   We respectfully refer to Sec. 4.2 in the manuscript. GAM-RAG remains lightweight in both the offline indexing stage and the online retrieval stage, with competitive token cost and latency.
>
> > ### **Regarding W2 and Q2 : Query Type and Query Ordering**
>
> **Query Type.** We respectfully refer to Supplementary Material Eq. (14), where the applicability of our memory mechanism is defined by a query class rather than repeated identical queries:
> $$
> Q_{\lambda}(q^{(\cdot)}) = \\{ q' \in \mathbb{R}^d \mid \|q'\|_2 = 1,\ \cos(q', q^{(\cdot)}) \ge \lambda \\}.
> $$
> Under this query-class assumption, our theory shows that the memory update is effective. Following the setup in Supplementary Material D, Table 2 in the main manuscript evaluates robustness under non-repetitive queries. We consider three scenarios: **Same Query**, **Similar Query** (paraphrased equivalents), and **Different Query** (same question type but different semantics).  The results show that GAM-RAG remains robust under both Similar Query and Different Query settings.
>
> **Query Ordering.** We fully agree that testing robustness to query ordering can further strengthen the evaluation. To this end, based on the same three settings above, we additionally conduct **10 shuffled runs** with different query orders and report the standard deviation of performance：
>
> Table 1: GPT-Acc (%) of different methods over 10 shuffled query runs.
>
> | Methods                          |       Same Query |    Similar Query |  Different Query |
> | -------------------------------- | ---------------: | ---------------: | ---------------: |
> | HippoRAG2                        |     55.68 ± 0.06 |     54.08 ± 0.07 |     54.06 ± 0.07 |
> | LinearRAG                        |     61.68 ± 0.03 |     60.28 ± 0.05 |     60.56 ± 0.06 |
> | REMINDRAG                        |     62.03 ± 0.18 |     61.47 ± 0.21 |     60.52 ± 0.25 |
> | GAM-RAG                          |     65.26 ± 0.15 |     64.50 ± 0.18 |     62.76 ± 0.22 |
> | GAM-RAG (-5-turn Memorization)   | **71.58 ± 0.37** | **70.82 ± 0.42** | **69.41 ± 0.50** |
>
> Overall, the variation across different orderings is small, indicating that our gains are not driven by a particular query sequence and that GAM-RAG is robust to query shuffling.
>
> > ### **Regarding W3 and Q1: Long-Term Stability Analysis**
>
> We respectfully refer to Supplementary Material A.1, Eq. (21):
>
> $$
> (1-\kappa(R))^n \le \frac{1-\lambda}{1-s_t}
> $$
>
> This indicates that the memory converges into the $\lambda$-similarity query cluster under consistent feedback.
>
> We acknowledge that noisy feedback is a common challenge for memory updating. In our setup, memory **confidence** increases only when a sentence is judged as supportive and the answer is correct. Thus, occasional wrong feedback may cause short-term bias, but the confidence-controlled gain limits its effect and helps prevent erroneous evidence from being easily solidified. In addition, the offset term $Q$ avoids confidence collapse and further mitigates error fixation.
>
> Besides, we further expanded the analysis of noisy feedback in the Sec. 4.3 (Long-term Stability) by randomly injecting feedback noise at rates of **10%** and **20%** .
>
> Table 2: GPT-Acc (%) under different noisy-feedback ratios across memory updates.
>
> | Noise Ratio |   R0 |   R1 |   R2 |   R3 |   R4 |   R5 |   R6 |   R7 |   R8 |   R9 |  R10 |
> | ----------- | ---: | ---: | ---: | ---: | ---: | ---: | ---: | ---: | ---: | ---: | ---: |
> | 0%          | 64.2 | 65.3 | 67.8 | 69.4 | 70.6 | 71.5 | 71.9 | 72.2 | 72.3 | 72.8 | 72.9 |
> | 10%         | 63.8 | 64.6 | 64.4 | 66.5 | 66.2 | 69.3 | 70.5 | 71.2 | 71.7 | 72.1 | 72.4 |
> | 20%         | 61.8 | 62.6 | 62.0 | 63.9 | 63.4 | 65.7 | 68.0 | 69.0 | 69.8 | 70.6 | 71.8 |
>
> As shown in table2, although noisy feedback causes larger fluctuations in the early stage, the performance gradually becomes stable as memory confidence increases.

---

> > ### Author Rebuttal · Reviewer_rmfz · 2026-04-03
> >
> > I appreciate the authors for the feedback and will maintain my score.

---

> > > ### Author Response · Authors · 2026-04-03
> > >
> > > Dear Reviewer rmfz,
> > >
> > > Thank you very much for your positive feedback and for your careful consideration of our rebuttal. We sincerely appreciate your time and support.
> > >
> > > Best regards,
> > >
> > > Authors of Paper34058

---

### Decision · Program_Chairs · 2026-04-30

**Decision:**

Accept (regular)

**Comment:**

This paper proposes GAM-RAG, a training-free evolving RAG framework that incrementally updates sentence-level memory and uncertainty so the system can accumulate retrieval experience across recurring or related queries, reducing repeated multi-hop retrieval while improving both accuracy and efficiency.

strengths:

* The problem is real and practically meaningful, since static RAG often repeats similar retrieval work for related queries.
* The method is fairly coherent, with the training-free design, Kalman-inspired gain rule, hierarchical retrieval, and memory update mechanism fitting together well.
* The empirical story is solid overall, with consistent accuracy gains together with clear reductions in retrieval cost and latency.

The main reviewer concerns focused on dependence on query ordering or repeated-query patterns, long-term drift, memory storage scalability, NER robustness, judge overhead, and the lack of clearer automatic metrics and terminology. The rebuttal addressed most concerns.
And reviewer  ot7E  still have the concerns for the novelty of this paper even they increase scores.
The remaining concerns feel more like caution about deployment scope than evidence against the method itself, so I give a weak accept for this paper.